# Tunable Mode Converter Device Based on Photonic Crystal Fiber with a Thermo-Responsive Liquid Crystal Core

**Jorge Andres Montoya Cardona** [1] , **Nelson Dario Gomez Cardona** [1],
**Esteban Gonzalez Valencia** [2], **Pedro Torres Trujillo** [2] **and Erick Reyes Vera** [1,*]

[1] Department of Electronic and Telecommunications Engineering, Instituto Tecnológico Metropolitano, Medellin A. A 050013, Colombia; jorgemontoya223139@correo.itm.edu.co (J.A.M.C.); nelsongomez@itm.edu.co (N.D.G.C.)

[2] Escuela de Física, Universidad Nacional de Colombia—Sede Medellín, Medellín A.A. 3840, Colombia; egonzalv@unal.edu.co (E.G.V.); pitorres@unal.edu.co (P.T.T.)

* Correspondence: erickreyes@itm.edu.co; Tel.: +57-4600727 (ext. 5586)

**Abstract:** A compact tunable mode converter device based on the thermo-optically characteristics of liquid crystals (LCs) is proposed and numerically analyzed herein. The proposed mode converter consists of an asymmetric dual-core photonic crystal fiber (PCF) with a highly thermo-responsive LC core. The verification of the proposed mode converter was ensured through an accurate PCF analysis based on the vector finite element method. With an appropriate choice of the design parameters associated with the LC core, phase matching at a single wavelength is available in the important O-band wavelength region. The simulation results showed that high conversion efficiencies between $LP_{01}$ and $LP_{11}$ mode are readily achieved over a broad wavelength range from 1278 nm to 1317 nm. Likewise, the tunable capability of the proposed mode converter was evaluated when it was submitted to thermal changes; thus, we evidence the strong thermo-responsive dependence of the operating wavelength, mode conversion efficiency and full-width at the half maximum (FWHM) bandwidth. Finally, the fabrication tolerances of the devices were also investigated. Therefore, the thermo-responsive characteristics of this novel PCF mode converter can be of fundamental importance in the future space division multiplexing technology.

**Keywords:** mode-division multiplexing; mode converter; photonic crystal fiber; tunable photonic device; liquid crystal; microstructured optical fiber

## 1. Introduction

In the last two decades, optical fibers have been the preferred transmission technology of metropolitan networks and long-haul terrestrial and transoceanic links, since they present several advantages, such as low-loss, large bandwidth and electromagnetic immunity, compared to other technologies. According recent reports, the traffic in optical communication networks has presented an exponential growth in the last decade as consequences of the rapid growth of internet users and the necessity to connect the users with objects [1–3]. To satisfy the demands of channels, more data have been transmitted over the same optical fiber by upgrading equipment at the optical fiber ends. Nevertheless, many experts around the world predict that the next decade or so, an increasing number of fibers in real networks will reach their capacity limits [3,4]. In order to increase the transmission rates and satisfy the growing demand for information, it is required that we develop several technological breakthroughs, such as low-loss single mode fibers (SMF) [5,6], erbium-doped

fiber amplifiers (EDFAs) [7,8], wavelength division multiplexing (WDM) [2,9], polarization division multiplexing (PDM) [10] and high-spectral-efficiency coding through a digital signal processor (DSP) to enable coherent transmission [11]. WDM is the dominant technology that consists of transmitting signals using each wavelength as an independent channel. It was developed to increase the fiber link capacities; however, it additionally has the advantages of easy routing and switching based on wavelength schemes [12]. Currently, WDM coherent optical communication systems use all the degrees of freedom, such as frequency, quadrature and polarization, in a single-mode fiber. Therefore, the only extra physical dimension is the space, which can be employed in the future optical fiber communication systems. In order to use this degree of freedom, space-division multiplexing (SDM) has been proposed [3], where independent data streams can be transmitted in parallel spatial channels. To perform SDM transmission, mode multiplexers (MUX) and demultiplexers (DEMUX) are critical components to transform signals from individual SMFs to SDM signals containing the superposition of the different modes. However, the implementation of this technique requires two important components: the first one is the mode converter, which takes the fundamental mode $LP_{01}$ and transforms it to a higher order mode such as $LP_{11}$ or $LP_{21}$. The second component is the combiner, which takes all propagating modes and combine them in a same physical channel. However, one of the main problems to implementing the SDM transmission technique has been the development of a transmission technique, since it requires the use of platforms based on optical fiber in order to reduce the insertion losses and integration with the current networks.

To solve this problem, several alternatives based on optical fiber technology had been explored in the past, such as FBGs, LPFGs, tapered fibers and applying lateral pressure [13–16]. Recently, schemes of mode conversion based on modal coupling in asymmetric dual-core optical fibers have been proposed [17–21]. Cheng et al. numerically analyzed the mode coupling between a large core and a small core in an asymmetric, dual-core photonic crystal fiber (PCF), reporting a $LP_{01}$ to $LP_{02}$ mode converter with a bandwidth of 14 nm and a length of 12.7 mm; it operates at around 1550 nm [17]. Later, Lin et al. presented a theoretical design of a $LP_{01}$ to $LP_{02}$ mode converter based on a special dual-core fiber for dispersion compensation by using the coupled-mode theory. They reported a photonic device with a 22 nm bandwidth and a coupling efficiency of up to 80% when operating at around 1550 nm. In addition, the simulation results showed that the conversion bandwidth can be improved to $\approx$31 nm by tapering the dual-core fiber [18]. Cai et al. proposed a scheme of mode conversion based on mode coupling in a hybrid dual-core PCF, which contains an index-guided core and a photonic bandgap core. Similar than other previous works, the analysis was made using numerical techniques; specifically, the beam propagation method (BPM). Likewise, the mode converter response is modulated by changing the refractive index (RI) of the hole between the two cores, being able to control bandwidth, operating wavelength and coupling efficiency thought variations of the RI of this hole [21]. Recently, Reyes-Vera et al. proposed a tunable mode converter based on a dual-core PCF filled with a thermo-responsive oil, which can operate in the S + C + L + U bands, and its performance can be controlled by thermal changes [22]. More recently, Yu et al. proposed a tunable magnetic fluid-filled hybrid PCF mode converter designed to convert the $LP_{11}$ mode in the index-guiding core to the $LP_{01}$ mode in the photonic bandgap-guiding core. They found that the coupling efficiency can reach 90% and 95% in the wavelength ranges 1.33–1.85 μm and 1.38–1.75 μm, respectively, with bandwidth values of 520 nm and 370 nm, in a 835 μm long fiber [23].

On the other hand, in the last decade, many researchers have explored the possibility of obtaining tunable photonic devices using hybrid PCFs, which offer new alternatives and flexibility to the optic communications systems [6,24,25] and sensing devices [26,27]. To develop this new generation of photonic devices based on hybrid PCFs, many alternatives have been explored, including PCFs filled with thermo-responsive materials such as metals, oils, polymers, semiconductors and liquid crystals (LCs), since their characteristics allow tuning the conditions of light propagation in these fibers [24,28–31]. In the above cases, some important things need taking into account: The first one is that the fluid-material absorption should be small at the wavelengths of operation. The second

one is that the fluid needs to have surface affinity with the fiber's host material to facilitate filling and avoid subsequent spilling of the fluid. Finally, the used material used to fill the PCFs require a high dependence of its optical properties with thermal changes; for most fluids the thermos-optic coefficient ($\partial n/\partial T$) is of the order of $\approx 10^{-6}\,\text{K}^{-1}$, but can be much larger for liquid crystals, which present a thermo-optic coefficient for LC materials of approximately $10^{-2}\,\text{K}^{-1}$, as reported by Gauzia et al. in [32]. In addition, LCs offer a better alternative since they are anisotropic and their RIs can be controlled by applying an external electric field or by changing the temperature. For those reasons, PCFs filled with LCs have been used for developing many tunable photonic devices; some of them were employed to design filters [33], polarization beam splitters (PBSs) [34,35], optical switches [36] and sensors [37]. For example, Saitoh et al. proposed a three-core PCF coupler with two identical cores separated by a third one that acts as a liquid crystal resonator. The novelty of this design is that the thermo-responsive characteristics of the LC were used to obtain a highly selective compact tunable bandpass filter, which offers a total temperature resolution of 2.5 nm/°C [33]. Cheng et al. presented the numerical study of a PBS based on dual-core silica glass PCF with a liquid crystal modulation core. The authors demonstrated the capability of the designed device to tune the operating wavelength. The temperature analysis evidenced that the proposed PBS covers the E + S + C + L optical communication bands, with an extinction ratio better than −20 dB. In addition, the PBS exhibits a satisfactory splitter performance when the fabrication tolerance reaches 1% [35].

PCF mode converters have been investigated in the C-band wavelength region; very few reports are available in the important O-band wavelength region. Recently, Younis et al. analyzed an asymmetric dual-core PCF wavelength-selective polarization splitter. The device performance was tuned by applying an external electric field, which was used to split out each of the polarization modes. The proposed photonic device reaches an excellent performance because it operates at 1.33 μm and 1.55 μm [34]. In this case, the operation band depends of the orientation of the LC molecule.

In this paper, we propose and numerically analyze a compact tunable PCF mode converter device based on the thermo-optically tunable characteristics of the LCs. By using a commercial numerical algorithm based on the vector finite element method, we demonstrated that in an asymmetric dual-core PCF with a highly thermo-responsive nematic LC core, high conversion efficiencies between $\text{LP}_{01}$ and $\text{LP}_{11}$ mode are achieved over a wavelength range from 1278 nm to 1317 nm. Furthermore, the performance degradation caused by fabrication error was analyzed, showing that this device has large fabrication tolerances. The thermo-responsive characteristics of this novel PCF mode converter can be of essential importance in the future SDM technology.

## 2. Materials and Methods

### 2.1. Structure Design

Figure 1 shows the cross section of the proposed mode converter based on a dual-core PCF, in which the two cores are separated by an air hole with a diameter ($d_a$) of 0.8 μm. This hole has an important role in the proposed design since its size, position and refractive index influence the mode coupling efficiency between the fiber cores, and the mode confinement. The left core consisted of a micro-hole of 1 μm in diameter ($d_{LC}$), infiltrated with a highly thermo-responsive nematic liquid crystal (NLC), E7 [38]. On the other side, the right core was a solid core of 4.2 μm in diameter ($d_{Ge}$), doped with germanium (concentration of 19.3 mole%). The Ge doping increased the refractive index of the solid core, allowing the interaction with the LC-core. As the RI of each core was higher than in the microstructured cladding (pure silica), the light-guiding mechanism in both cases was based on total internal reflection (TIR). The other holes that form the hexagonal microstructure had a diameter ($d_h$) of 2 μm, and the structure pitch ($\Lambda$) was 3.5 μm. The design parameters of the microstructure were selected in order to reduce the losses in the photonic device. Here, the losses are minimized as consequence of the presence of the air-holes microstructure and the high RI contrast between the cores and the background material. The RI of the background silica and the doped core were modeled using

the Sellmeier's equation, and the coefficients were taken from [2]. On the other hand, the E7 NLC RI has a strong dependence on the light wavelength and the applied temperature. Therefore, numerical calculations of the device characteristics require considering the impact of both parameters on the material dispersion. The wavelength and thermo-optical dispersion properties for LC molecules were considered based on the Cauchy coefficients reported by Li et al. [38]:

$$n_{\text{e}}(T,\ \lambda) \approx A_e + \frac{B_e}{\lambda^2} + \frac{C_e}{\lambda^4}, \tag{1a}$$

$$n_{\text{o}}(T,\ \lambda) \approx A_o + \frac{B_o}{\lambda^2} + \frac{C_o}{\lambda^4}, \tag{1b}$$

where $A_o$, $B_o$, $C_o$, $A_e$, $B_e$ and $C_e$ are the Cauchy coefficients as functions of the applied temperature [38]. As an example, the Cauchy coefficients at room temperature (25 °C) are $Ao = 1.4994$, $Bo = 0.0070$ $\mu m^2$, $Co = 0.0004\ \mu m^4$, $Ae = 1.6933$, $Be = 0.0078\ \mu m^2$ and $Ce = 0.0028\ \mu m^4$; therefore, the ordinary and extraordinary refractive indexes are 1.503614 and 1.698795 at 1310 nm, respectively. Now, using the ordinary and extraordinary RI components, the relative permittivity tensor can be obtained, which describes the optical behavior of the LC in each direction. This tensor is given by

$$\varepsilon_r = \begin{pmatrix} n_o^2 sin^2\theta + n_e^2 cos^2\theta & \left(n_e^2 - n_o^2\right)sin\theta cos\theta & 0 \\ \left(n_e^2 - n_o^2\right)sin\theta cos\theta & n_e^2 sin^2\theta + n_o^2 cos^2\theta & 0 \\ 0 & 0 & n_o^2 \end{pmatrix}, \tag{2}$$

where $\theta$ is the rotation angle of the E7 NLC molecules with respect the x-axis. The relative permittivity tensor shows that the liquid crystal is anisotropic, and each component has a strong dependence on the orientation of the E7 NLC molecules.

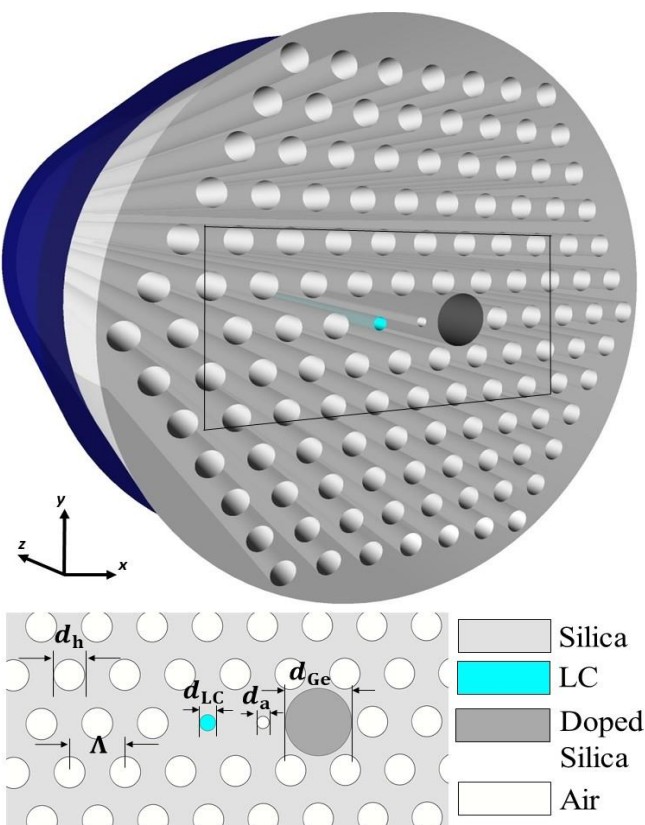

**Figure 1.** Schematic of the tunable photonic crystal fiber (PCF) mode converter with a thermo-responsive liquid crystal (LC) core.

Silica PCFs have been realized with various fabrication techniques, including the well-known stack and draw method [39,40] and the extrusion method [41]. Any of the techniques are suitable for the proposed design because it is based only on circular holes, which makes its fabrication easier compared to structures with elliptical holes or more complex shapes [41]. In the past, several PCFs with holes of diameter 0.8 μm or less were fabricated. For example, Knight et al. studied the group-velocity dispersion in several air-silica PCFs [42]. They made the measurements using PCFs with diameters of air holes of 0.4 μm, 0.62 μm, 1.1 μm and 1.24 μm. Then, they demonstrated the strong dependence of the group-velocity dispersion with the diameter of holes and other geometrical parameters in this type of optical fiber. Zhou et al. reported the fabrication of a glass PCF employing a die-cast process in 2006 [43]. So, they reported a PCF with hexagonal shape and the diameter of air holes in clad, core, and pitch of PCF as 0.8 μm, 1.7 μm and 1.3 μm, respectively. One year after, Wiederhecker et al. reported the fabrication and characterization of a PCF with a subwavelength-scale air hole [44]. In fact, the hole had a diameter of 0.2 μm or less. Therefore, they demonstrated that a hole with subwavelength-scale helped to confine the light. Finally, Ebendorff-Heidepriem et al. showed that using the extrusion method, it is possible to obtain PCFs whose structures present high quality. Actually, they reported a PCF with circular holes close to the core with diameters of 0.9 μm [41]. On the other hand, after the fabrication of the PCF structure, the hole with diameter $d_{LC}$ should be infiltrated with NLC to establish the second core [45,46]. Some techniques have been explored for the micro-hole selective filling in PCF structures with NLC and other liquid materials [30,47,48]. For example, Vieweg et al. employed the two-photon direct-laser writing technique to seal individual holes with SU-8 photoresist and the subsequent selective filling of PCF holes with highly nonlinear liquids and LCs [49]. Then, this technique could be used to fill the LC-core of the proposed structure.

## 2.2. Working Principle

The operating principle of the modal converter is based on the coupled mode theory (CMT) [50], for which we can consider two different cases. In the first one, both cores are identical, and the effective RI of each core matches the other one; thus, the coupling between them is perfect since the phase-matching condition is always achieved. In the second case, both cores are non-identical, whereby the transfer of power between the two cores only occurs at the phase-matching wavelength. Therefore, the power is transferred periodically between both cores, and the maximum power transfer occurs only at the coupling length ($L_C$). However, a weak coupling can occur away from phase-matching wavelength despite the absence of phase matching.

As shown in Figure 1, the proposed mode converter device was based on the second case, since it is an asymmetric structure and the cores are made of different materials. Therefore, to begin this study, the behavior of the dispersion curves associated with each core has to be evaluated in order to detect the phase-matching condition and evidence the influence of the polarization effects. Then the CMT can be used, analyzing each core independently, and the interactions between them can be considered a perturbation to the whole structure. As the fiber is not absorbent, the electric field of the system can be treated as a linear superposition of the individual core fields as

$$E(x, y, z) = A_1(z)e^{-i\beta_1 z}E_1(x, y) + A_2(z)e^{-i\beta_2 z}E_2(x, y).$$ (3)

The subscripts 1 and 2 refer to the Ge-doped and LC core, respectively; $E_{1,2}$, $A_{1,2}$ and $\beta_{1,2}$ are the electric field distributions, amplitudes and propagation constants of the propagating modes in each isolated core, respectively. The coupled mode equation can be written as

$$\frac{d}{dz}\begin{bmatrix} A_1 \\ A_2 \end{bmatrix} = i\begin{bmatrix} \beta_1 & \kappa_{12} \\ \kappa_{21} & \beta_2 \end{bmatrix}\begin{bmatrix} A_1 \\ A_2 \end{bmatrix},$$ (4)

where $\kappa_{12}$ are z-dependent the coupling coefficients defined as [51]

$$\kappa_{jl} = \frac{k_0}{2n_{co,j}} \frac{\int_A (n^2 - n_l^2) E_j^* \cdot E_l dA}{\int_A E_j^* \cdot E_j dA} \tag{5}$$

with $(j,l) = (1,2)$ or $(2,1)$, for which the longitudinal field components have been ignored [51]. In Equation (5) $k_0$ is the vacuum wave number, $n$ is the refractive index profile of the whole system, $n_l$ is the refractive index profile of the system for the each isolated *l*-th core, $n_{co,j}$ is the refractive index of the *j*-th core and $A$ is the PCF cross-section. In a step-index dual-core fiber, the coupling coefficients can be analytically calculated using Equation (4); however, in a dual core PCF a numerical model, such as the vector finite elements method (FEM) should be used.

The linear system of Equation (3) is solved assuming that the light is injected into the *LC*-core (left core) that only transmits the fundamental mode (LP$_{01}$), which is coupled through the evanescent field during light propagation to the LP$_{11}$ mode of the Ge-doped core (right core). Based on this computational methodology, one can obtain the effective RI associated to each mode and its electric field distribution. Next, the coupling coefficients could be obtained under different conditions by using Equation (5).

The cross section of the optical fiber under study has a circular geometry with a radius of 22.75 μm. The maximum element size in mesh is set five times smaller than the working wavelength. A cylindrical perfectly matched layer (PML) boundary condition, with a thickness of 1.55 μm was used to absorb the scattered electromagnetic waves towards the surface [52]. With these conditions, the total number of domain elements in the mesh was 68,278 with 4,530 boundary elements. It is worth mentioning that the PML thickness and element size have great impacts on the simulation results.

## 3. Results and Discussion

To begin the analysis of the proposed mode converter, the modal dispersion curves of the propagating modes in each core were calculated in the wavelength range 1280–1320 nm. This analysis was carried out at room temperature for two different rotation angles of the E7-LC molecules, $\theta = 0°$ and $\theta = 90°$, and using the geometrical parameters described in Section 2.1. Figure 2a,b shows the dispersion characteristics of the *x*- and *y*-polarized modes. Figure 2a shows the modal dispersion curves of the *x*-polarized propagating modes of the Ge-doped core (LP$_{01x}$, LP$_{11ax}$ and LP$_{11bx}$) and LP$_{01}$ modes of the *LC*-core. Here, it is important to precise that LP$_{01,o}$ and LP$_{01,e}$ was obtained when the LC E7 molecules were oriented with θ equal to 90° (dashed black line) and θ equal to 0° (red line) respectively. The modal dispersion curves obtained allow us to evidence that when the light is injected parallel to *x*-axis, there is only a coupling between the LP$_{01,o}$ mode and the LP$_{11ax}$ mode, as evidenced by the crossing point at $\lambda_o = 1295.2$ nm, while other modes they do not show a phase matching condition in the spectral range studied. Similarly, Figure 2b shows the modal dispersion curves of the *y*-polarized propagating modes of the Ge-doped core (LP$_{01y}$, LP$_{11ay}$ and LP$_{11by}$) and the LP$_{01}$ modes of the *LC*-core, where the LP$_{01,o}$ and LP$_{01,e}$ modes were obtained as before, when the E7-LC molecules were oriented at $\theta = 0°$ (black line) and at $\theta = 90°$ (dashed red line) respectively. From this figure, one can observe a similar result; i.e., when the light is injected parallel to *y*-axis, there is only a coupling between LP$_{01,o}$ mode and the LP$_{11ay}$ mode, but then at $\lambda_o = 1306.4$ nm. Based on these results, the mode converter device could be operating at any of the polarization conditions in which its performance depends on the light polarization.

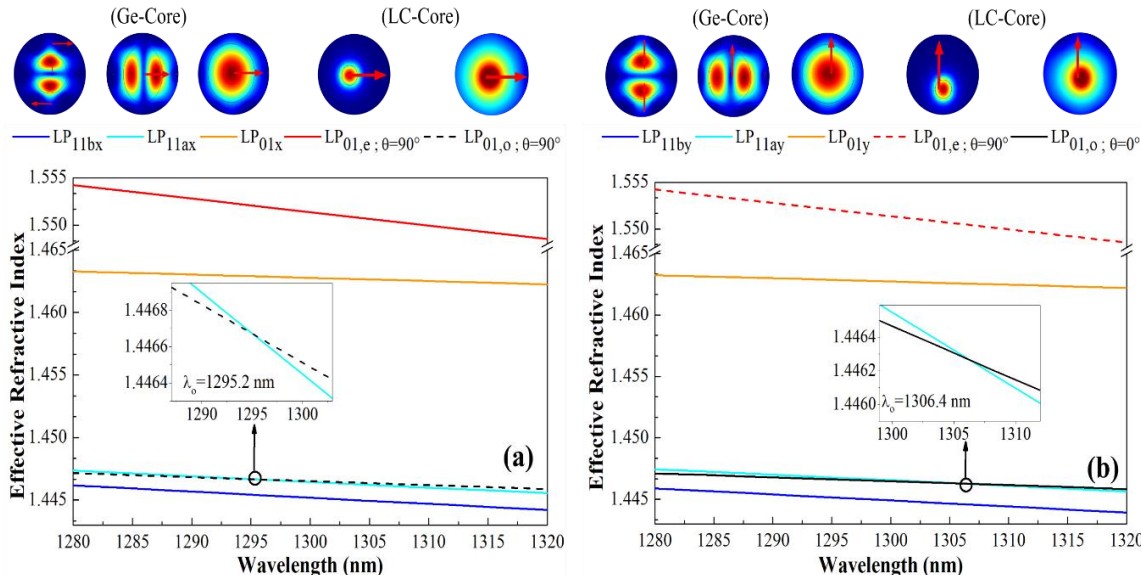

**Figure 2.** Modal dispersion curves of the proposed mode converter device at 25 °C for (**a**) *x*-polarized modes and (**b**) *y*-polarized modes. The red and black dashed lines indicate the $LP_{01}$ modes of the LC-core at $\theta = 90°$, while the red and black solid lines indicate the $LP_{01}$ modes of the LC-core at $\theta = 0°$. The insets illustrate the electric field distribution of each propagating mode in the structure.

To analyze the tuning capability of the proposed mode converter, the modal dispersion curves of both cores at different temperatures were calculated. As an example, Figure 3a shows the dispersion curves at $\theta = 0°$ in the temperature range from 15 to 35 °C. It can be seen that for the left core, the effective RI of the $LP_{11ay}$ mode decreases linearly as the wavelength increases and fundamentally does not change with the temperature due to the low dependence of the Ge-doped silica RI with the temperature, which facilitates the coupling with the $LP_{01,o}$ mode of the right core. On the other hand, the dispersion curve of the thermo-responsive LC core mode presents a displacement due to the fact that the temperature changes the LC RI [38]. Therefore, the phase matching condition, and hence the wavelength to which the device operates, can be thermally tuned.

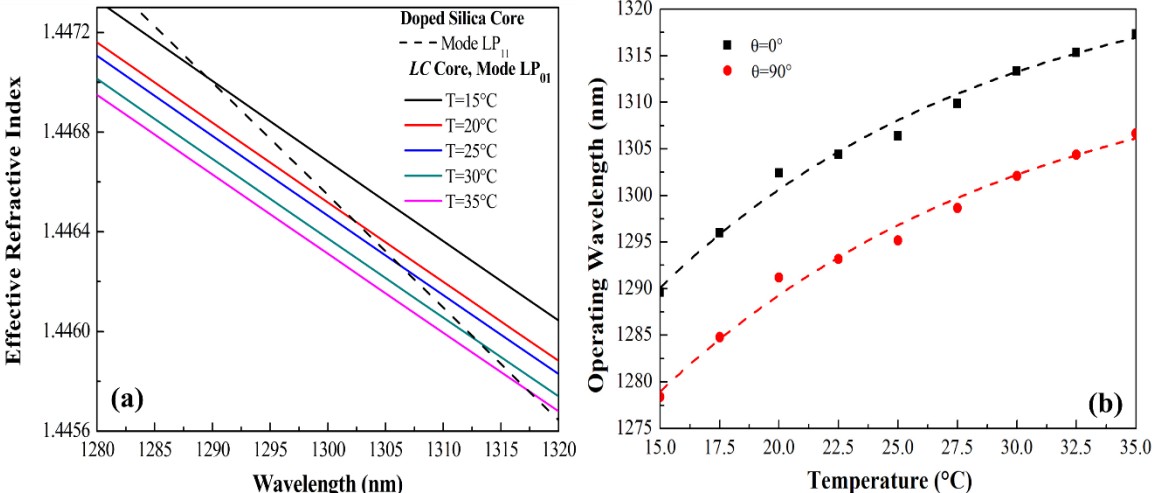

**Figure 3.** (**a**) Modal dispersion curves of the $LP_{01,o}$ mode in the LC-core at $\theta = 0°$ and $LP_{11ay}$ mode in the Ge doped core at different temperatures. (**b**) Operating wavelength against temperature.

The results with different orientation angles of the LC molecule evidence that the phase-matching wavelength could be tuned through temperature variations. In fact, for $\theta = 0°$ and $\theta = 90°$ the operating

wavelength of these proposed mode converter device is tunable and has a nonlinear dependence on temperature as illustrated in Figure 3b. We note that the Equations (6a) and (6b) provide a good description of the operating wavelength as a function of the applied temperature—a mode analysis, not included in this work, taking into account the non-linear temperature dependence of the LC RI [26] is quite satisfactory. The results in Figure 3b evidence that the tunable mode converter can operate from 1289.6 nm to 1317.3 nm with $\theta = 0°$ and from 1278.4 nm to 1306.66 nm with $\theta = 90°$ in the thermal range analyzed, which indicates that the device has tunabilities ($\Delta\lambda/\Delta T$) of 1.385 nm/°C and 1.413 nm/°C, respectively. At this point it is worth saying that, evidently, the mode converter analysis was repeated for the two rotation angles of the E7 LC; from now on only the case $\theta = 0°$ will be mentioned.

$$\lambda_o = -102.56641 exp\left(-\frac{T}{14.10066}\right) + 1325.501; \ \theta = 0° \tag{6a}$$

$$\lambda_o = -98.98757 exp\left(-\frac{T}{15.38025}\right) + 1316.297; \ \theta = 90°. \tag{6b}$$

The mode coupling efficiency has a strong dependence on both the wavelength and the length of the mode converter [51]. Figure 4a shows the mode coupling efficiency as a function of the length of the mode converter device at different temperatures for a fixed wavelength of 1310 nm. From these results, the mode converter achieves the best performance when the device has a length of approximately 3.15 mm, since at this length the mode converter reaches a good efficiency in most cases, showing only a poor response at T = 15 °C. As can be seen in this figure, the mode converter reaches a maximum mode conversion efficiency of 99% at 30 °C. This is because at this temperature, the phase-matching point is close to the desirable operating wavelength, as is clear from Figure 3b. In addition, the results show that the mode coupling efficiency has values above 60% when the temperatures is between 20 and 35 °C, which means that the proposed mode converter could be implemented into O-band with an excellent performance.

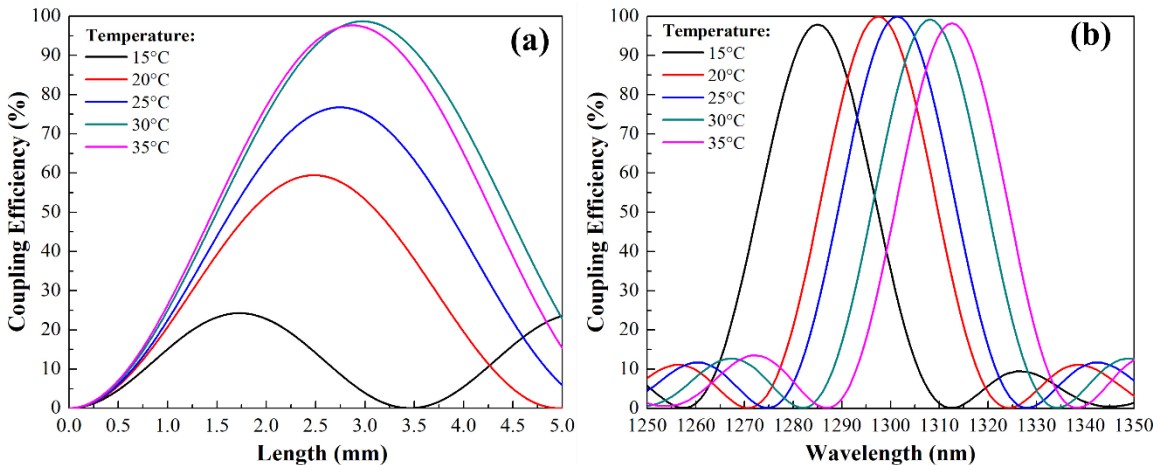

**Figure 4.** (**a**) Mode coupling efficiency between the $LP_{01,o}$ mode (left core) and $LP_{11ay}$ (right core) along the propagation direction for a fixed wavelength of 1310 nm and $\theta = 0°$. (**b**) Mode coupling efficiency against wavelength at different temperatures for a 3.15 mm long $LP_{01,o}$–$LP_{11ay}$ mode converter.

Once the optimal length of the mode converter device was determined, the next step was to study the spectral performance of the proposed device at different temperatures. Figure 4b shows the mode coupling efficiency for a device with an optimal length of 3.15 mm. From this figure, the proposed mode converter operates in the O-band (1260 nm to 1360 nm) and the operating band could be tuned by small thermal changes. Moreover, the coupling efficiency of this mode converter reaches a maximum value of about 99% in the phase-matching wavelength. In addition, the spectral band of the mode coupling efficiency maintains a similar shape and only changes to a longer wavelength when the

temperature increases; it is a consequence of the phase-matching wavelength shift, which ensures that this device has high efficiencies in the range of temperatures studied.

The mode converter bandwidth is another important parameter since it indicates the capability of this device to be used in combination with the WDM technique simultaneously, or it could be employed as a wavelength selective coupler. The bandwidth of the proposed device was measured at the full-width at the half maximum (FWHM). Therefore, the dependence of the FWHM bandwidth on temperature was also investigated. From Figure 4b, the FWHM bandwidth of $LP_{01}$-$LP_{11}$ mode converter changes slightly with temperature. For example, the FWHM bandwidths at 15, 20, 25, 30 and 35 °C are, respectively, 19.9375, 19.4147, 19.2959, 18.9926 and 18.6982 nm. Table 1 shows a comparison between the proposed mode converters with existing structures. The comparison has in account many important factors, such as structure type, tunable capability, operating range, bandwidth and total length of device. From this table, it is evident that the proposed mode converter presents a good performance compared with previous reports and represents the first thermo-responsive mode converter using optical fiber technology.

**Table 1.** Comparison among the properties of the proposed mode converter with previously reported works.

| Description | Tunable | Range | Bandwidth | Length | Ref. |
|---|---|---|---|---|---|
| Mode converted based on mode coupling in an asymmetric dual-core PCF | No | 1550 nm | 14 nm | 12.7 mm | [17] |
| Mode converter based on dual-core all-solid PCF | No | 1550 nm | 47.8 nm (max.) | 6.437 mm | [20] |
| Mode Converter Based on Polymer Waveguide Grating | Yes | 1560 nm to 1592 nm | 4 nm | 5.07 mm | [53] |
| Hybrid dual-core PCF | No | 1550 nm | 43 nm (max.) | 3.21 mm | [21] |
| Adiabatically tapered MOF mode converter | No | 1550 nm | NP [1] | 21 mm | [54] |
| Mode converter based on the LPFG written in the two-mode fiber | Yes | 1500 nm to 1540 nm (max.) | 18 nm (max.) | 24 mm | [55] |
| Dual core Hollow-Core PBGF | No | 1965 nm | 200 nm | 7.6 mm | [56] |
| Magnetic Fluid-Filled Hybrid Connect Dual-Core PCF mode converter | Yes | 1.33 μm–1.85 μm 1.38 μm–1.75 μm | 0.52 μm 0.37 μm | 0.835 mm | [23] |
| Tunable mode converter device based on PCF with a thermo-responsive LC-core | Yes | 1278 nm to 1317 nm | 19.938 nm | 3.15 mm | This work |

[1] The authors do not provide this information.

## 4. Fabrication Tolerance Analysis

The geometric parameters determine the way in which the light propagates in the fiber structure; moreover, they have important influences on the optical properties of the excited modes in both cores, and, consequently, on the coupling efficiency. In this section, we evaluate the behavior of the mode coupling efficiency when the inner lattice hole diameter ($d_h$), the inner diameter of hole between cores ($d_a$), the inner diameter of LC filled hole ($d_{LC}$) and the diameter of $GeO_2$ doped core ($d_{Ge}$). They are varied individually while the length of the device remains constant at the optimal value. Furthermore, computational simulations are performed for tolerances of 2% and 4% of the values of the geometric parameters indicated above and at a temperature of 25 °C.

Lattice hole diameter plays an important role in the propagation parameters of a PCF, mainly because it determines the behavior of the effective RI of the propagating modes. In this mode coupler, the amount of transferred power between cores depends of phase matching of non-identical cores, a

condition that depends on $d_h$. As shown in Figure 5a, the phase-matching wavelength is modified according to the deviation in the nominal value of $d_h$, as a result of the changes that obviously occur in the effective RIs of the propagating modes involved in the mode conversion scheme. Contrary to this situation, Figure 5c shows that the deviation in the nominal value of $d_a$ slightly affects the phase-matching wavelength because this parameter does not strongly affect the effective RIs of the propagating modes, and, therefore, the phase matching condition is not significantly affected. In both cases, there are no appreciable effects on the mode coupling efficiency, which indicates that variations in these parameters do not appreciably affect the coupling coefficient $\kappa_{jl}$ in Equation (5).

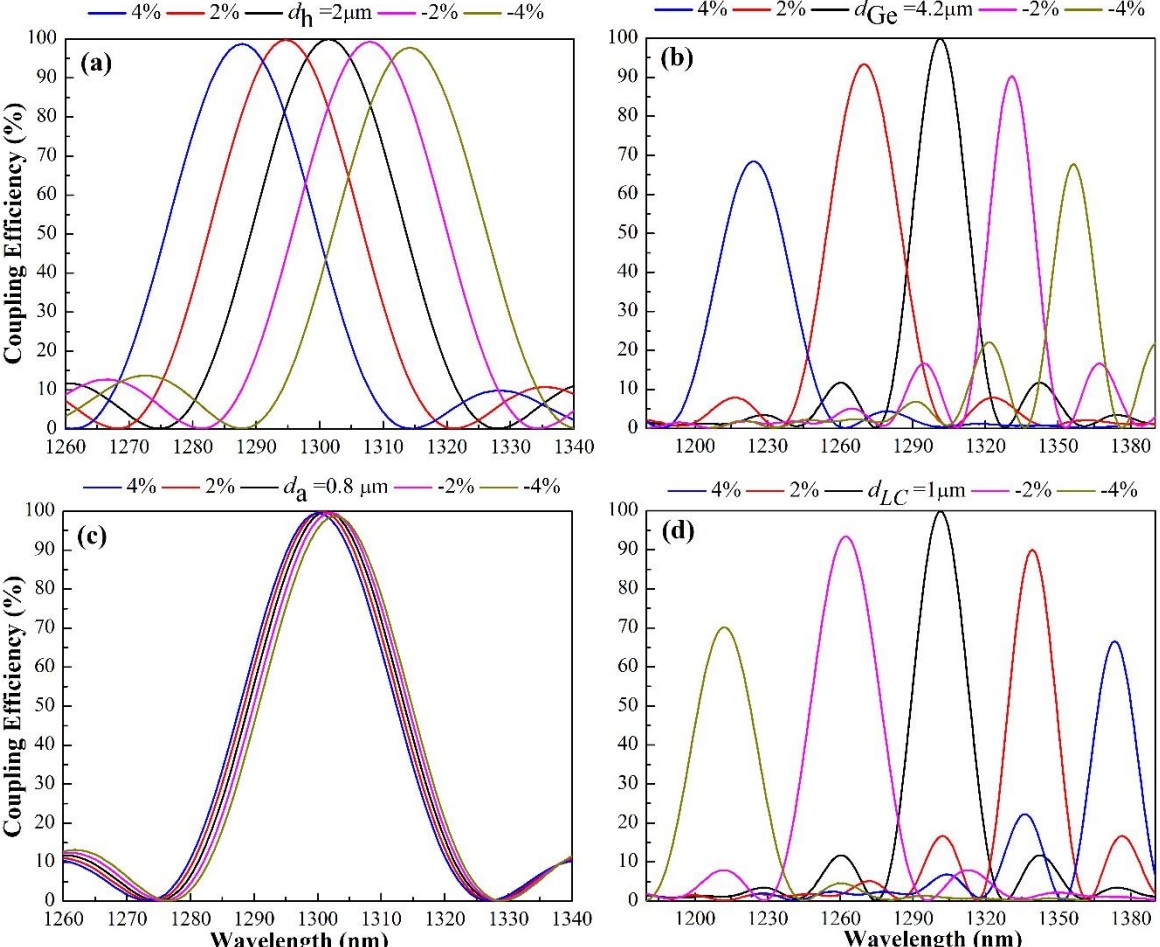

**Figure 5.** Mode coupling efficiency of the 3.15 mm long $LP_{01,o}$–$LP_{11ay}$ mode coupler considering variations in (**a**) lattice hole diameter (dh), (**b**) GeO2 doped core diameter (dGe), (**c**) diameter of the hole between cores (da) and (**d**) LC filled hole diameter (dLC).

Figure 5b,d shows the behavior of mode coupling efficiency when the diameter of a Ge-doped core and LC-filled hole are modified, respectively. Unlike to the variations of $d_h$ and $d_a$, which affect the whole structure, variations on $d_{Ge}$ and $d_{LC,}$ affect primarily the propagation characteristics of corresponding core, which leads to changes in the mode coupling efficiency. The aforementioned occurs mainly due to the change in the diameter of any of these cores, which affects the spatial distributions of their electric fields, and, consequently, the coupling coefficients in Equation (5). Obviously, by modifying these diameters, the phase-matching wavelength is also modified. For example, when $d_{Ge}$ is reduced, the phase-matching wavelength shifts to long wavelengths, while when $d_{LC}$ is reduced the phase-matching wavelength shifts to short wavelengths. In both cases, the mode coupling efficiency

is affected, because the evanescent field involved in the mode conversion scheme changes and the separation distance between cores increases, whereby the coupling coefficient is affected.

Figure 6a shows the dependence of mode coupling efficiency and FWHM bandwidth on fabrication tolerance. From these results, it is clear that the parameters that modify the structure as a whole do not significantly affect either coupling efficiency or FWHM bandwidth. On the other hand, as variations in the diameters of the $GeO_2$ doped core and the LC-filled hole modify the propagation properties of their respective cores, mode coupling efficiency and FWHM bandwidth are affected. Finally, Figure 6b shows the mode coupling efficiency as a function of wavelength for a tolerance of 5% in the device length. There are no changes in the phase-matching wavelength because this parameter does not affect the effective refractive index of the cores. Small changes in the maximum value of coupling efficiency and FWHM bandwidth are induced by slight variations in the power transferred from one core to another.

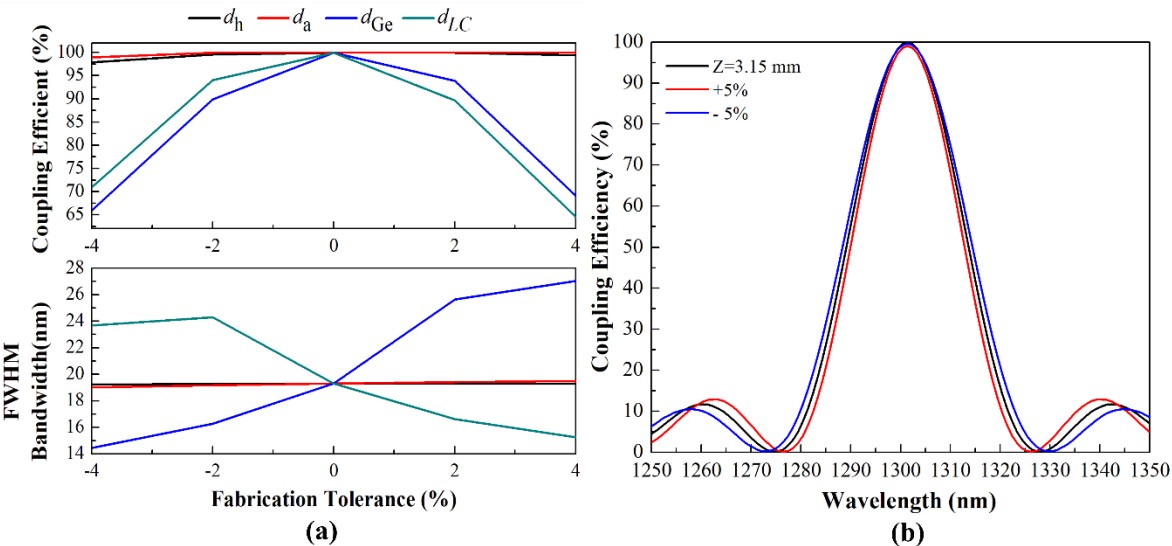

**Figure 6.** (**a**) Coupling efficiency and FWHM bandwidth of the $LP_{01,o}$–$LP_{11ay}$ mode converter as a function of fabrication tolerance. (**b**) Coupling efficiency against wavelength for a tolerance of 5% in the device length.

## 5. Conclusions

In conclusion, a compact tunable mode converter device based on mode coupling in an asymmetric dual core PCF with a highly thermo-responsive LC core was proposed and numerically analyzed. The thermal tuning capability of the phase matching condition was evaluated when the mode conversions were performed between $LP_{01}$ and $LP_{11}$ modes. We have shown that it is possible to control the operating wavelength of mode converter thought the thermo-optically tunable characteristics of the LCs. The proposed devices show high conversion efficiency over a broad wavelength range from 1278 to 1317 nm. The mode converter can be readily fabricated by PCF processing and has high fabrication tolerance.

**Author Contributions:** The work described in this article was the collaborative development of all authors. Conceptualization, E.R.V. and P.T.T.; methodology, E.R.V., E.G.V. and N.D.G.C.; software, J.A.M.C.; data curation, J.A.M.C. and E.G.V.; investigation, E.R.-V., N.D.G.C., E.G.V., P.T.T. and J.A.M.C.; resources, N.D.G.C.; writing—original draft preparation, E.R.V., E.G.V. and N.D.G.C.; writing—review and editing, E.R.V., P.T.T., E.G.V. and N.D.G.C.; supervision, P.T.T. All authors have read and agreed to the published version of the manuscript.

**Funding:** The authors would like to thank the support provided to this work by the Instituto Tecnológico Metropolitano, project P17217.

**Conflicts of Interest:** The authors declare no conflict of interest.

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
