# Peer review of "Tunable Mode Converter Device Based on Photonic Crystal Fiber with a Thermo-Responsive Liquid Crystal Core"

_photonics, doi:10.3390/photonics7010003_

Round 1

Reviewer 1 Report

The paper " Tunable mode converter device based on photonic crystal fiber with a thermo-responsive liquid crystal core" by J. Montoya-Cardona, et al., reports their design and numerical study on a compact tunable mode converter device. They explain how important this device can improve the transmission capacity in optical fibers. The authors of this paper explored the design by simulation, which is important for understanding the coupling mechanism and also how precise should be the device in terms of fabrication.

The paper is well written and the figures are clear. However, I believe the authors can be improved it more making some of the following changes.

Some comments:

The author does not specify the type of liquid crystal (LC) that they used for the analysis. Also, how the changes in the temperature for the LC-core for rotation of the molecules is affecting the Ge-doped core. Line 30: reference [1] is not updated. A current study should be included. Line 33-35: when the authors talk about the technological breakthroughs none of them are referenced. Each technology should be referenced to the original paper. Line 211: the use of the word “fundamental”, when refers to modes is confusing. It can be improved with propagation modes or inner modes. Line 214: There is a typo. The word “buy” should be changed by “but”. Line 217: At the top of figure 2 is hard to read the subscripts in which mode is plotted. Line 275: the bandwidth was measured at the full-width at the half maximum (FWHM)? Or at the half-width at 1/e of the maximum (HW1/eM)?

In terms of fabrication, the process is really complex. Although the geometry is circular, the liquid crystal selective infiltration is a hard task.  

I found this work appropriate for publication in Photonics.

Author Response

We are very much thankful to the referees for their deep and thorough review. We have revised our present research paper in the light of their useful suggestions and comments. We hope our revision has improved the paper to a level of their satisfaction. Please see the attachment.

Reviewer 2 Report

See the review report attached

Author Response

(The authors gave the same response as above.)

Reviewer 3 Report

Dear editor,

In this paper, the authors proposed and numerically analyzed a tunable mode converter device based on a thermo-optical LCs. The work lacks some interesting points. Similar works have been reported and their results are more convince. So I suggest authors conclude some novelty about the tunable device. In addition, I have some issue as follows.

Why authors not mention the tunable characteristics in the abstract? The authors conclude a broad wavelength in the range of 1278 nm to 1317 nm. Any potential application? Although authors present a numerical method they merely imply the simulated parameters.

Author Response

(The authors gave the same response as above.)
